# Variable-Length Tokenization via Learnable Global Merging for Diffusion Transformers

**Dong Hoon Lee** [1]  **Seunghoon Hong** [2]

## Abstract

Latent Diffusion Models (LDMs) have become dominant in visual synthesis, but their quality–compute trade-off is largely constrained by the tokenizer's fixed compression ratio. Variable-length tokenizers (VLTs) promise adaptive compression by varying token counts, allowing diffusion models to flexibly balance quality and compute. However, conventional VLTs modulate length by truncating ordered token sequences, which makes token semantics depend on token position and breaks representational alignment across lengths. This leads to a cross-length shift in the latent distribution that hinders a single variable-length diffusion model from operating effectively. To address this, we propose a novel variable-length tokenizer that modulates length by merging tokens. We show that encouraging similar tokens to merge enables direct cross-length representation alignment when the diffusion transformer operates according to the merging pattern. Since conventional merging methods are data-dependent, making the merging pattern inaccessible during generation, we introduce learnable global merging, which is data-independent, to ensure compatibility with diffusion transformers. On ImageNet $256 \times 256$ generation, our merging-based variable-length tokenizer integrated with a diffusion transformer achieves a superior gFID–compute trade-off compared to prior VLT methods.

## 1. Introduction

Latent Diffusion Models (LDMs) (Rombach et al., 2022) have emerged as a standard framework for visual synthesis, with their key breakthrough being the use of tokenizers to enable diffusion models to operate in compressed latent spaces. This reduces computational requirements while achieving superior generation quality compared to pixel-space diffusion models. Within this framework, the tokenizer's compression ratio becomes a critical factor (Rombach et al., 2022; Yu et al., 2024) shaping the quality-compute trade-off, where higher compression ratio dramatically reduces computational requirements but degrades generation quality and vice versa. This trade-off becomes challenging when dealing with diverse use cases, from high-fidelity applications that prioritize visual quality to resource-constrained scenarios that require aggressive compression for feasibility. Since conventional tokenizers employ fixed compression ratios, they cannot adapt to these varying requirements within a single model, necessitating multiple model variants for different computational budgets and quality targets.

In this pursuit, variable-length tokenizers (Miwa et al., 2025; Wen et al., 2025; Bachmann et al., 2025; Duggal et al., 2025) have emerged, which can adjust the compression ratio by varying the number of tokens used to represent data. The core idea is to introduce a length modulation mechanism that adjusts the number of tokens based on specific conditions such as computational budget or image complexity. The most prevalent mechanism is nested dropout (Miwa et al., 2025; Wen et al., 2025; Bachmann et al., 2025), which randomly drops tail tokens in a nested manner during training. This encourages an ordered token (Kusupati et al., 2022) sequence that places important information in early tokens, where drops are rare, allowing the prefix tokens to be used for reduced token counts. Paired with models (*e.g.*, AR models) that generalize across varying token lengths, these tokenizers promise flexible quality-compute trade-offs.

However, the conventional approach with nested dropout induces a cross-length shift in the datapoint-wise similarity structure. While some shift in similarity structure is inevitable due to changes in token length, nested dropout exacerbates this cross-length shift by truncating an ordered token sequence—where early tokens encode high-level semantics and later tokens store lower-level details. Truncating this sequence from the tail results in shorter sequences focusing on core semantics, while longer sequences emphasize residual details. Consequently, the pairwise relationships among

[1]Kim Jaechul Graduate School of AI, KAIST, Daejeon, South Korea [2]School of Computing, KAIST, Daejeon, South Korea. Correspondence to: Seunghoon Hong <seunghoon.hong@kaist.ac.kr>.

*Proceedings of the 43rd International Conference on Machine Learning*, Seoul, South Korea. PMLR 306, 2026. Copyright 2026 by the author(s).

datapoints differ across token lengths. This becomes particularly problematic for diffusion models, as it means the latent space distribution (and corresponding score function) varies significantly across token lengths, making it difficult to train a diffusion model that generalizes across varying token lengths for flexible quality-compute trade-offs.

This motivates us to encourage *representational alignment* (Huh et al., 2024; Sucholutsky et al., 2025) , *i.e.*, consistency of similarity structure across token lengths. While a common approach is to directly minimize the difference between representations (Yao et al., 2025; Yu et al., 2025; Leng et al., 2025), this is only feasible when representation dimensions (*i.e.*, token counts) can be matched to define the distance. This raises the need for a solution tailored to VLTs.

To this end, we propose merging-based variable-length tokenization as a solution for representational alignment. In our design, when length modulation is performed via merging and the diffusion model operates according to the merging patterns (*i.e.*, which tokens are combined), the merging pattern defines a full-length-equivalent representation of the modulated latents (with matched cardinality). This makes the representation shift induced by length modulation directly measurable (Eq. 5 in Sec. 3.1), and such shift can be minimized by combining similar tokens. As a result, in our framework, combining similar tokens leads to a reduction in representation shift, which subsequently encourages representational alignment.

One challenge is that conventional merging methods (Bolya et al., 2023; Lee & Hong, 2024; Haurum et al., 2024) cannot be directly used in our generative setting. Diffusion models must be aware of the merging pattern, but data-dependent methods derive it from the input image, which is unavailable at generation time. To address this, we introduce learnable global merging, where token merging is independent of the data and thus the merging patterns are accessible to generative models at generation time. Despite being data-independent, our learnable global merging is optimized to reconstruct images while combining similar tokens. This is achieved via a straight-through trick applied to learnable global embeddings that determine the merging pattern. We empirically verify that our global merging approach is sufficient for achieving strong reconstruction quality while preserving the similarity structure.

Finally, we train diffusion transformers across variable-length latents from our tokenizer to enable effective quality-compute control. We introduce merged positional embeddings to handle positional information of merged tokens jointly with proportional attention. Experiments on ImageNet 256×256 generation demonstrate that our tokenizer achieves improved gFID-compute trade-offs compared to existing VLTs. We also investigate lightweight length-specific

LoRA post-training to further improve the trade-off with negligible training/parameter overhead. Our contributions are summarized as follows:

- We propose a merging-based variable-length tokenizer that modulates length through token merging. Our tokenizer encourages consistent datapoint-wise similarity structure across token lengths, which facilitates diffusion training over different token counts.

- Experiments on ImageNet 256×256 generation demonstrate that our tokenizer, combined with a diffusion transformer, achieves the best trade-off between gFID and computational cost compared to existing variable-length tokenization methods.

## 2. Backgrounds

Latent Diffusion Models (LDMs) have become a standard approach in image generation, operating on a latent space that is learned through tokenization. The generation quality and computational cost of LDMs heavily depend on the structure of the latent space and the compression ratio of the tokenizer, respectively.

### 2.1. Image Tokenization

Image tokenization aims to map high-dimensional images $x \in \mathbb{R}^{H \times W \times 3}$ to compact latents $z \in \mathbb{R}^{N \times D}$, thereby reducing computational complexity for downstream models that operate on these latents, such as generative models or vision-language models. The tokenizer follows an encoder-decoder framework and is trained to reconstruct the original image. It consists of an encoder $z = \mathcal{E}(x)$ that encodes the image $x$ into latent tokens $z$ and a decoder $\hat{x} = \mathcal{D}(z)$ that reconstructs the image $x$. The training objective typically follows (Esser et al., 2021; Rombach et al., 2022):

$$\mathcal{L}_{\text{total}} = \mathcal{L}_{\text{rec}} + \lambda_{\text{per}}\mathcal{L}_{\text{per}} + \lambda_{\text{adv}}\mathcal{L}_{\text{adv}} + \lambda_{\text{reg}}\mathcal{L}_{\text{reg}}, \quad (1)$$

where the reconstruction loss $\mathcal{L}_{\text{rec}}$ enforces pixel-level reconstruction, the perceptual loss $\mathcal{L}_{\text{per}}$ and adversarial loss $\mathcal{L}_{\text{adv}}$ enhance perceptual quality (Esser et al., 2021), and the regularization loss $\mathcal{L}_{\text{reg}}$ prevents a high-variance latent space (Rombach et al., 2022; Chen et al., 2025b). Here, the $\lambda_{(\cdot)}$ terms denote the corresponding loss weights.

Note that conventional tokenizers (Esser et al., 2021; Rombach et al., 2022; Chen et al., 2025b) are limited to encoding at a single compression rate, *i.e.*, a fixed $N$ in $z \in \mathbb{R}^{N \times D}$. These tokenizers cannot adapt to varying computational budgets or quality requirements across different scenarios. This necessitates multiple tokenizers operating across different scenarios with varying compression rates, increasing development and deployment costs.

## 2.2. Variable-Length Tokenization (VLT)

Variable-length tokenization (Miwa et al., 2025; Duggal et al., 2025; Bachmann et al., 2025; Wen et al., 2025) presents a more challenging objective: to enable adaptive tokenization at multiple compression rates (*i.e.*, varying token counts). This allows adjusting the compression rate based on computational budget or task requirements, enabling flexible and efficient use of downstream models (Duggal et al., 2025; Bachmann et al., 2025). To achieve this flexibility, a core component is the length modulation mechanism that typically operates between the encoder and decoder to convert the full latent representation $z \in \mathbb{R}^{N \times D}$ into reduced modulated tokens $\tilde{z} \in \mathbb{R}^{M \times D}$ where $M < N$. The decoder then reconstructs the image from the modulated tokens $\tilde{z}$ as $\hat{x} = \mathcal{D}(\tilde{z})$, and the encoder and decoder are trained to reconstruct the image across varying token counts.

**Nested Dropout**   In particular, existing variable-length tokenization methods (Miwa et al., 2025; Wen et al., 2025; Bachmann et al., 2025) predominantly employ *nested dropout* to enable length modulation. Following the principle of Matryoshka representation learning (Kusupati et al., 2022), nested dropout encourages ordered representations by randomly dropping tail tokens during training. Specifically, given the full latents $z = (z_1, ..., z_N) \in \mathbb{R}^{N \times D}$, nested dropout randomly samples the number of tokens $K \in \{1, ..., N\}$ to remove and applies truncation to $z$:

$$\tilde{z} = \text{Trunc}[z, K] = (z_1, z_2, ..., z_{N-K}) \in \mathbb{R}^{(N-K) \times D}.$$

Here, the resulting token count is $M = N - K$. This encourages ordered representations by placing important high-level semantic information early in the sequence where drops rarely occur, while relegating low-level details to later positions that are more frequently dropped. After training, these tokenizers enable flexible length modulation by truncating low-level details from the tail.

## 2.3. Limitations of VLTs in Diffusion Models

Integrating variable-length tokenization with latent diffusion models presents a promising direction. By training a single variable-length diffusion model to operate effectively across diverse token lengths, this integration enables flexible generation of high-quality images under varying computational budgets. However, despite its promise, the compatibility of variable-length tokenization methods with latent diffusion models poses challenges, particularly due to the effect of nested dropout on *representational alignment* (Huh et al., 2024).

**Limitations**   However, nested dropout's ordered representation can be limiting for variable-length latent diffusion models: truncating the tail shifts the datapoint-wise similar-

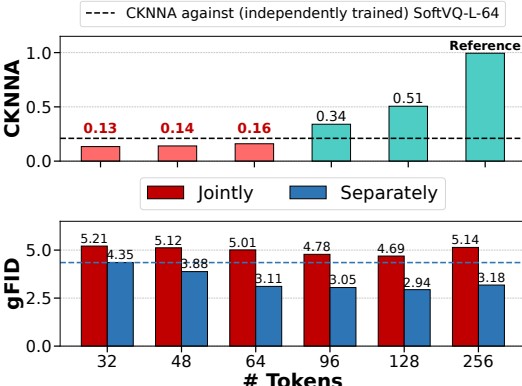

*Figure 1.* **CKNNA and gFID measurements from the nested dropout tokenizer and the diffusion model trained on it, respectively**. *Upper*: CKNNA between untruncated ($N = 256$) and truncated ($M < 256$) latents; the black dotted line shows CKNNA for an independently trained tokenizer (vs. 256). *Lower*: gFID of the variable-length diffusion model learned jointly across token lengths vs. separately for each length.

ity structure across token lengths, which weakens representational alignment (Huh et al., 2024; Sucholutsky et al., 2025) — the consistency of pairwise relationships (*i.e.*, similarity structure) among latent representations across different token lengths. Specifically, as discussed, nested dropout encodes high-level semantic content in early tokens while relegating residual low-level details to later positions. Consequently, when the token length $M$ varies, the modulated tokens $\tilde{z}$ exhibit a cross-length shift in the similarity structure among datapoints in latent space. Such similarity structure drift leads to significant variation in the latent distribution (and thus the score function that diffusion models must learn) across different token lengths $M$, making it difficult to train a single diffusion model that generalizes well across token lengths.

As shown in Fig. 1, we measure representational alignment (*i.e.*, consistency in similarity structure) using Centered Kernel Nearest-Neighbor Alignment (CKNNA), and observe that nested dropout latents exhibit markedly lower cross-length consistency—often even lower than CKNNA observed between independently trained fixed-rate tokenizers. Consistent with this, training a single variable-length diffusion model on nested dropout latents jointly across token lengths leads to gFID degradation compared to training length-specific diffusion models. This undermines the benefit of flexible computation: even with more tokens, the jointly trained model fails to outperform a model trained solely on fewer tokens, as shown by the blue dotted line.

A common approach to encouraging representational alignment is to use representation-alignment losses (Yao et al., 2025; Yu et al., 2025) that minimize the difference between representations[1], thereby promoting consistent similarity

---

[1]distance$(z, \tilde{z}) < \delta$

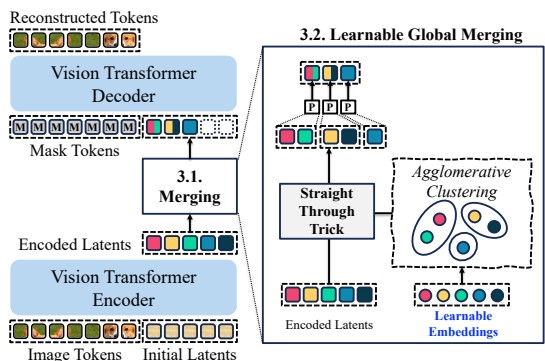

*Figure 2.* **Overview of our method.** To encourage representational alignment, we propose length modulation via merging (Sec. 3.1). The merging operation is image-agnostic (*i.e.*, globally determined) but optimized for tokenization and alignment (Sec. 3.2).

structure. However, these losses commonly require the two representations to have the same number of tokens ($N = M$), which does not hold across variable-length tokens. While alignment metrics for different cardinalities exist, they either involve discrete operations such as nearest neighbor search within a large batch (*e.g.*, CKNNA), making direct optimization challenging, or capture only global statistics while neglecting local similarity structure among individual latents (*e.g.*, aligning mean-pooled tokens), providing limited benefit for diffusion models that must model the entire latent distribution. These challenges suggest that maintaining consistent similarity structure across token lengths requires a different strategy.

## 3. Method

Our objective is to design a variable-length tokenizer that encourages representational alignment (*i.e.*, consistency in similarity structure) across token lengths, facilitating the training of variable-length diffusion models. To this end, we propose to merge tokens to modulate token length, which makes direct alignment of latents across token lengths possible. We first explain how we merge tokens to modulate token length and why this enables direct alignment of latents across token lengths, despite differences in cardinality (Sec. 3.1). We then introduce our approach for learning the merging pattern while keeping it compatible with diffusion-based generation (Sec. 3.2). Finally, we present our training recipe for variable-length diffusion transformers (Sec. 3.3). Figure 2 provides an overview of our approach.

### 3.1. Merging-based Length Modulation

We begin by explaining our length modulation via token merging, assuming the merging pattern (*i.e.*, which tokens to merge) is given as an assignment matrix $\Gamma \in \{0,1\}^{N \times (N-K)}$, where $\Gamma_{ij} = 1$ indicates that the $i$-th token is assigned to the $j$-th cluster. Given $\Gamma$, the length-

modulated latents $\tilde{z} \in \mathbb{R}^{(N-K) \times D}$ are computed as:

$$\tilde{z} = \bar{\Gamma}^{\mathsf{T}} z, \quad \text{where} \quad \bar{\Gamma}_{ij} = \frac{\Gamma_{ij}}{\sum_k \Gamma_{kj}}, \quad (2)$$

yielding $N - K$ merged tokens each representing the cluster center of its connected original tokens. We define the merged token sizes $m \in \mathbb{R}^{N-K}$, which count the number of original tokens each merged token of $\tilde{z}$ contains, as $m_j = \sum_i \Gamma_{ij}$.

Once length-modulated latents $\tilde{z}$ are obtained, we use them to reconstruct the original image with the decoder or to synthesize images with diffusion transformers (DiTs). To do so, we employ *proportional attention* in attention modules of the decoder and DiTs, as proposed in (Bolya et al., 2023), to account for each merged token's effective cluster size $m_j$ (*i.e.*, how many original tokens it aggregates). Specifically, given token sizes $m \in \mathbb{R}^{N-K}$, the self-attention score is modified as $A = \text{Softmax}\left(\frac{QK^{\mathsf{T}}}{\sqrt{d}} + \log m\right)$. Importantly, computing $N - K$ merged tokens $\tilde{z}$ with proportional attention is equivalent to processing the projected latents $Wz \in \mathbb{R}^{N \times D}$ with the standard attention, where $W \in \mathbb{R}^{N \times N}$ denotes the projection matrix defined to replace each token with the average of all tokens in the same cluster:

$$W_{ij} = \begin{cases} \frac{1}{|\mathcal{C}_i|} & \text{if } j \in \mathcal{C}_i \\ 0 & \text{otherwise} \end{cases}, \quad (3)$$

where $\mathcal{C}_i = \{j : \exists c, \Gamma_{ic} = 1 \text{ and } \Gamma_{jc} = 1\}$ denotes the set of tokens assigned to the same cluster as token $i$. Note that $Wz$ has $N - K$ distinct rows, which correspond to the $N - K$ merged tokens of $\tilde{z}$, each repeated by its effective size $|\mathcal{C}_i|$. Consequently, processing the projected latents $Wz$ with standard self-attention is equivalent to computing proportional attention over the modulated latents $\tilde{z}$.[2]

The equivalence under proportional attention thus defines full-length equivalent latents $Wz \in \mathbb{R}^{N \times D}$ for any modulated latents $\tilde{z} \in \mathbb{R}^{(N-K) \times D}$ regardless of token length $N - K$. Since both the original latents $z$ and the full-length equivalent latents $Wz$ share the same cardinality $N$, we can directly align latents by minimizing their difference, *i.e.*, $\|Wz - z\|^2$, as in representation alignment losses in (Yao et al., 2025; Yu et al., 2025). Notably, $\|Wz - z\|^2$ corresponds to:

$$\|Wz - z\|^2 = \sum_{i=1}^{N} \|z_i - (Wz)_i\|^2 \quad (4)$$

$$= \frac{1}{2} \sum_{i=1}^{N} \frac{1}{|\mathcal{C}_i|} \sum_{j \in \mathcal{C}_i} \|z_i - z_j\|^2, \quad (5)$$

and is thus minimized when merging groups similar tokens within $z = (z_1, \ldots, z_N)$. As a result, with merging-based

---

[2]See Bolya et al. (2023) for details.

length modulation, we can directly align latents across token lengths, despite differences in cardinality, by encouraging (token-wise) similar tokens within each sequence $z$ to merge.

### 3.2. Learnable Global Merging

Note that in terms of merging (*i.e.*, which tokens to merge), Eq. 5 can be minimized by applying clustering to latents $z$. Unfortunately, such $z$- or $x$-dependent merging is incompatible with image generation: it requires an image-specific assignment $\Gamma$ that is unknown before the image is generated. Specifically, since generation does not have access to the target image (or its latent $z$), the merged token sizes $m$ and merged positions cannot be computed. This prevents the application of proportional attention during generation, making operations on modulated latents $\tilde{z}$ no longer equivalent to those on full-length equivalent latents $Wz$. Similarly, assigning proper positional embeddings requires knowing which tokens are combined, which is also unavailable during generation. This constrains the use of conventional token merging (Bolya et al., 2023; Haurum et al., 2024) or clustering methods based on $z$ (or $x$) in our merging-based modulation.

Therefore, we propose a learnable global merging approach where merging is determined independently of $z$ but optimized for tokenization and alignment of latents. Such independence from $z$ allows the merged token sizes $m$ and merged positions to be consistent across different token lengths, enabling compatibility with diffusion models via proportional attention and proper positional embeddings. The merging is learned to minimize the tokenizer training objective jointly with alignment loss, which encourages similar tokens to merge. While this approach does not provide an optimal solution to representation alignment in Eq. 5, we empirically show that it still achieves competitive reconstruction quality while improving cross-length similarity-structure consistency.

**Learnable Global Merging**  Specifically, we introduce learnable embeddings $e = \{e_1, e_2, ..., e_N\} \in \mathbb{R}^{N \times D}$ where each embedding $e_i$ corresponds to the $i$-th position of the latent token to determine which tokens to combine for each target reduction $K$. Given $K$, we apply agglomerative (hierarchical) clustering (Haurum et al., 2024) to form $N - K$ clusters based on cosine similarity between the embeddings in $e$, resulting in an assignment matrix $\Gamma$, where $\Gamma_{ij} = 1$ indicates that the $i$-th embedding $e_i$ is assigned to the $j$-th agglomerative cluster:

$$\Gamma = \text{Agglomerative}(e) \in \{0,1\}^{N \times (N-K)}, \quad (6)$$

$$\tilde{z} = \bar{\Gamma}^{\mathsf{T}} z \quad \text{where} \quad \bar{\Gamma}_{ij} = \frac{\Gamma_{ij}}{\sum_k \Gamma_{kj}}. \quad (7)$$

Note that agglomerative clustering comprises discrete $\arg\min$ operations that block gradients, preventing direct optimization of the learnable embeddings $e$ through gradient descent. To address this, we apply a straight-through trick (Bengio et al., 2013) to $\Gamma$, enabling the training of learnable embeddings:

$$\text{cluster centers: } c = \bar{\Gamma}^{\mathsf{T}} e, \quad (8)$$

$$\text{soft assignments: } \Gamma^{\text{soft}} = \text{softmax}(\frac{ec^{\mathsf{T}}}{\tau}), \quad (9)$$

$$\text{straight-through trick: } \Gamma \to [\Gamma - \Gamma^{\text{soft}}]_{\text{sg}} + \Gamma^{\text{soft}}, \quad (10)$$

where $[\cdot]_{\text{sg}}$ denotes the stop-gradient operation and $\tau$ is a scaling parameter. By applying the straight-through trick, we can optimize the assignment matrix determined by learnable embeddings $e$ using:

$$\tilde{z} = ([\Gamma - \Gamma^{\text{soft}}]_{\text{sg}} + \Gamma^{\text{soft}})^{\mathsf{T}} z \quad (11)$$

as the modulated latents. The decoder then reconstructs the image from the modulated latents $\tilde{z}$ as $\hat{x} = \mathcal{D}(\tilde{z})$, and the learnable embeddings $e$ are trained via the tokenizer's training objective $\mathcal{L}_{\text{total}}$ (Eq. 1) and the alignment loss $\mathcal{L}_{\text{align}}$, which encourages similar tokens to merge.

**Alignment loss**  Note that similar tokens $z_i, z_j$ in latents $z$ are merged only when their corresponding embeddings $e_i, e_j$ in learnable embeddings $e$ are also similar.[3] To encourage similar tokens to merge, we train the learnable embeddings to reflect the *token-wise* similarity between tokens in latents $z$ via an alignment loss, as in Yao et al. (2025). Specifically, given the full encoded latents $z$ for each image $x$, the alignment loss is defined as:

$$\mathcal{L}_{\text{align}} = \sum_{i,j} \text{ReLU}\left(\left|\left[\frac{z_i \cdot z_j}{\|z_i\|\|z_j\|}\right]_{\text{sg}} - \frac{e_i \cdot e_j}{\|e_i\|\|e_j\|}\right| - \delta\right),$$

where $\delta$ is a margin to prevent over-regularization. This alignment loss encourages the token-wise similarity between each pair of latent tokens $z_i, z_j$ to align with the similarity between their corresponding learnable embeddings $e_i, e_j$, thereby guiding the global merging to group similar tokens and minimizing Eq. 5.

Finally, our overall loss for global merging becomes:

$$\min_{e, \mathcal{E}, \mathcal{D}} \mathcal{L}_{\text{total}} + \lambda_{\text{align}} \mathcal{L}_{\text{align}}. \quad (12)$$

This objective is intended to preserve reconstruction quality while encouraging representational alignment across token lengths.

---

[3]Similarity here refers to *token-wise* similarity within a single latent sequence $z$, not the cross-datapoint similarity structure discussed earlier.

### 3.3. Application to Diffusion Transformers

We train a diffusion transformer (DiT) across variable-length latents from our tokenizer to enable controllable generation quality-compute trade-offs. Specifically, we simply train the diffusion transformer $G_\psi$ with a standard diffusion (or velocity) loss across varying-length latents $\tilde{z}$ by randomly sampling token lengths $K$:

$$\mathcal{L}_{\text{diff}}(\psi) = \mathbb{E}_{K,\tilde{z}}||G_\psi(\tilde{z}_t, t) - \epsilon_t||^2, \qquad (13)$$

where $G_\psi$ predicts the noise $\epsilon_t$ added to the merged latents $\tilde{z}$ of length $N - K$. As explained in Sec. 3.1, we use proportional attention (Bolya et al., 2023) in DiT to account for the effective size $m$ of merged tokens.

Unlike conventional DiTs operating on fixed-length tokens, our DiT must encode the positional information of merged tokens that aggregate original tokens from multiple positions. We find that merging positional embeddings in the same manner as the latents $z$ is sufficient in practice. Specifically, given learnable positional embeddings $p_e \in \mathbb{R}^{N \times D}$ and the assignment matrix $\Gamma$ for a token length of $N - K$, the positional embeddings for the diffusion model input become:

$$\tilde{p}_e = \bar{\Gamma}^\intercal p_e. \qquad (14)$$

We also experimented with separate positional embeddings for each token count but observed negligible improvement over the merged approach.

Additionally, while native DiT training with the merged positional embedding strategy alone achieves reasonable performance, we find that a simple post-training procedure introducing length-specific LoRA parameters for each token count can further improve results. Specifically, after joint training, we introduce length-specific LoRAs and train them for a small number of additional epochs (*i.e.*, 10 epochs). This allows the DiT to adapt more closely to each token count, providing consistent performance gains at the cost of negligible training/parameter overheads[4].

## 4. Experiments

**Dataset and Metrics**   Following the variable-length tokenization baselines (Miwa et al., 2025; Wen et al., 2025), we primarily conduct experiments on class-conditional image generation on ImageNet-1k (Deng et al., 2009) at $256 \times 256$ resolution. For evaluating image generation quality, we report the Fréchet Inception Distance for generation (gFID) (Heusel et al., 2017) with classifier-free guidance (Ho & Salimans, 2021) applied. For computational cost, we report the overall FLOP counts (TFLOPs)

---

[4]Each token length adds only 2.5% training overhead and 2.4–3.4% LoRA parameters.

and throughput (images/s) of the image generation pipeline. In our analysis (Sec. 4.3), we also report Peak Signal-to-Noise Ratio (PSNR), Structural Similarity Index Measure (SSIM), and the reconstruction Fréchet Inception Distance (rFID) (Heusel et al., 2017) on the ImageNet-1k validation split to evaluate the reconstruction quality of the tokenizers.

**Tokenizer**   We adopt a 1D tokenizer architecture, following the variable-length tokenization baselines (Miwa et al., 2025; Wen et al., 2025; Bachmann et al., 2025). Specifically, we implement our variable-length tokenizer on top of the SoftVQ (Chen et al., 2025b) framework, which consists of a Vision Transformer (ViT) encoder and a ViT decoder. The tokenizer uses a patch size of 16 and a latent dimension of 32, with a maximum token length of 256. We train a ViT-B-based 1D tokenizer with a batch size of 256 for 25 epochs. Images are center-cropped without any augmentation except for horizontal flipping and normalized to $[-1, 1]$. The tokenizer is trained with a combination of reconstruction, perceptual, adversarial, regularization, and alignment losses. For perceptual loss, we employ a pretrained VGG (Simonyan & Zisserman, 2015) network, and for adversarial loss, we use a pretrained DINOv1 (Caron et al., 2021) ViT-S encoder with a StyleGAN (Karras et al., 2019) discriminator operating on top of it. We apply LeCAM regularization (Tseng et al., 2021) for discriminator training while keeping the pretrained encoder weights frozen. For the regularization loss, we combine KL regularization $\mathcal{L}_{\text{kl}}$ from SoftVQ with semantic regularization $\mathcal{L}_{\text{SR}}$, where a ViT-Tiny auxiliary decoder is trained to predict semantic representations from a pretrained DINOv2 (Oquab et al., 2024) ViT-L encoder, following Chen et al. (2025a), resulting in $\mathcal{L}_{\text{reg}} = \mathcal{L}_{\text{kl}} + \mathcal{L}_{\text{SR}}$. We provide further architectural details in the Appendix.

**Diffusion Transformer**   We train LightningDiT (Yao et al., 2025) models on top of our latents to evaluate variable-length diffusion generation. We follow the training recipe and hyperparameters from LightningDiT (Yao et al., 2025), using a velocity prediction loss across varying-length latents. Specifically, we train LightningDiT models with a patch size of 1 and a global batch size of 1024, using AdamW with a learning rate of $2 \times 10^{-4}$ and without warm-up or weight decay. We also adopt SwiGLU FFN and RMSNorm, while replacing RoPE with 1D learnable positional embeddings to support 1D tokenization.

**Baselines**   We compare our tokenizer against both variable-length and high-compression fixed-length 1D tokenizers: (1) Semanticist (Wen et al., 2025) and FlexTok (Bachmann et al., 2025), variable-length tokenizers that enable flexible computational costs for image generation, and (2) SoftVQ (Chen et al., 2025b) and MAETok (Chen et al., 2025a), fixed-length tokenizers that achieve strong perfor-

*Table 1.* **Conditional generation quality (gFID) with different length modulations across varying numbers of tokens.**

| Modulation | Joint | # Tokens | | | | |
|---|---|---|---|---|---|---|
| | | 32 | 48 | 64 | 96 | 128 |
| Nested Dropout | ✗ | 4.35 | - | 3.11 | - | 2.94 |
| | ✓ | 5.21 | 5.12 | 5.01 | 4.78 | 4.69 |
| Ours | ✗ | **2.99** | - | **2.80** | - | **2.48** |
| | ✓ | 3.19 | 3.11 | 3.09 | 2.87 | 2.79 |
| + LoRA | - | 3.04 | 2.92 | 2.78 | 2.72 | 2.52 |

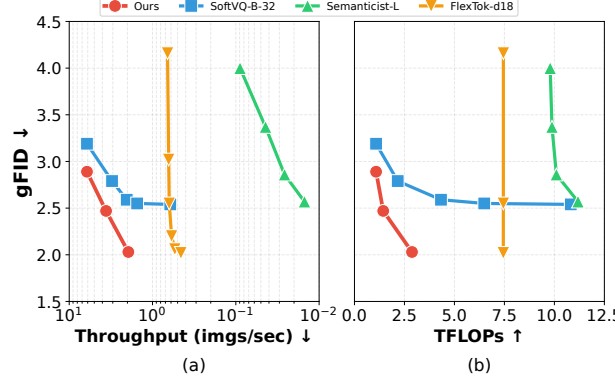

*Figure 4.* **Image generation results under different FLOPs.** The computational cost includes all FLOPs from (1) the generative model and (2) tokenizer decoding. In this comparison, our method uses the 25-step sampling configuration.

**Representational Alignment**   Figure 3 presents CKNNA scores measuring cross-length representational alignment. Our method achieves substantially higher cross-length CKNNA scores compared to nested dropout (0.76 vs. 0.34 on average), showing that our alignment objective effectively encourages consistent representations. Notably, our method consistently yields higher CKNNA scores than independently trained tokenizers, whereas nested dropout does not, verifying the encouraged representational alignment.

**Generation under Joint Training**   Table 1 presents the conditional image generation performance of diffusion models across varying token lengths. Note that nested dropout suffers from a significant gFID gap between jointly trained and separately trained models (0.8–1.7), likely due to inconsistent similarity structures across lengths. In contrast, our method achieves a smaller gap (<0.3), demonstrating improved generalizability of diffusion models across token lengths. Moreover, when applying LoRA fine-tuning as a post-training step, our approach achieves performance within 0.05 gFID of separately trained models with a minimal additional parameters and training. These results demonstrate that combining learnable global merging with diffusion across token lengths enables an effective quality-compute trade-off.

*Figure 3.* **CKNNA between encoded tokens with different token counts.** Each subplot indicates the reference token length (256, 128, 64, 32), and data points in the line plot indicate the CKNNA measured for each token count on the x-axis. The horizontal line indicates the CKNNA measured against an independently trained fixed-length tokenizer that shares the same tokenizer architecture.

mance at high compression rates (≤ 128 tokens). Since existing VLTs differ in tokenizer architectures and training objectives, direct comparisons may reflect both length modulation and other design choices. We therefore additionally re-implement One-D-Piece (Miwa et al., 2025) using our SoftVQ-based architecture to provide a controlled comparison under the same tokenizer backbone.

## 4.1. Impact of Learnable Global Merging

We first evaluate whether Learnable Global Merging improves cross-length representational alignment and joint variable-length diffusion training. To analyze the impact of our Learnable Global Merging as a replacement for nested dropout, we train tokenizers with identical architectures while varying only the length modulation module. We then train LightningDiT-B (Yao et al., 2025) models on these latent spaces under two settings: (1) joint, where a single model operates across all token lengths, and (2) independent, where a separate model is trained per length.

## 4.2. Comparison to Baselines

We next compare our method with existing VLT and fixed-length tokenizer baselines to evaluate the quality-compute trade-off.

**Comparison to Variable-Length Tokenizers**   Figure 4 presents image generation results under varying FLOPs and throughputs compared to other variable-length tokenization methods applicable to generative models, including Semanticist (Wen et al., 2025) and FlexTok (Bachmann et al., 2025). As an additional baseline, we include a fixed-length

*Table 2.* **System-level comparison of tokenization and generation pipelines on ImageNet** $256 \times 256$ **conditional generation**. The sampling-step setting for our method is indicated in the method name.

| Method | Type (T) | # Params (T) | Model (G) | Type (G) | # Params (G) | # Tokens | rFID | gFID | IS | TFlops | im/s |
|---|---|---|---|---|---|---|---|---|---|---|---|
| *2D tokenizers* | | | | | | | | | | | |
| SD-VAE | CNN | 84M | DiT-XL/2
SiT-XL/2
+ REPA | Diffusion | 675M | 1024
(256) | 0.62 | 2.27
2.06
1.42 | 278.2
270.3
305.7 | -
-
- | -
-
- |
| VA-VAE | CNN | 70M | LightningDiT-XL | Diffusion | 675M | 256 | 0.28 | 1.35 | 295.3 | - | - |
| *1D tokenizers* | | | | | | | | | | | |
| TiTok-B-64
One-D-Piece-B-256* | ViT | 176M | MaskGIT-ViT | Masked | 177M | 64
256 | 1.70
1.11 | 2.48
2.70 | 216.6
259.3 | -
- | -
- |
| SoftVQ-B-32
MAETok-B-128 | ViT | 176M | SiT-XL
LightningDiT-XL | Diffusion | 675M | 32
128 | 0.89
0.48 | 2.51
1.73 | 301.3
308.4 | 10.82
- | 0.61
- |
| *Variable-length tokenizers* | | | | | | | | | | | |
| Semanticist-L
Semanticist-XL | Diffusion | 609M
827M | $\epsilon$LlamaGen-L | MAR | 489M | 32
32 | 1.68
1.40 | 2.57
2.57 | 260.9
254.0 | 11.18
16.09 | 0.015
- |
| FlexTok d18-d18
FlexTok d18-d28 | Diffusion | 573M
1.4B | Custom AR | AR | 1.33B | 32
32 | 1.61
1.45 | 2.02
1.86 | -
- | 7.45
27.52 | 0.46
- |
| Ours (25 steps) | ViT | 176M | LightningDiT-XL | Diffusion | 675M | 32
128 | 1.37
0.59 | 2.89
2.03 | 311.2
297.4 | 1.08
2.88 | 6.12
1.95 |
| Ours (100 steps) | ViT | 176M | LightningDiT-XL | Diffusion | 675M | 32
128 | 1.37
0.59 | 2.54
1.86 | 302.3
290.7 | 4.33
11.51 | 2.04
0.36 |

*\* One-D-Piece supports variable-length tokenization, but its generator is not jointly trained across lengths.*

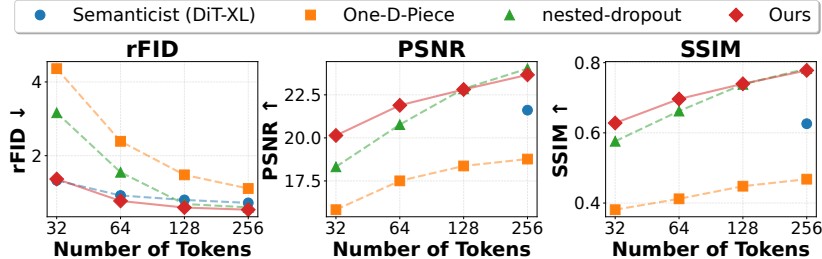

*Figure 5.* **Comparison on reconstruction quality with varying token lengths.**

*Table 3.* **Ablation study on our method.** The first row (all ✗) denotes token merging (Bolya et al., 2023) without proportional attention or positional embeddings applied to the DiT-B.

| Components | | | gFID / # Tokens | | | | |
|---|---|---|---|---|---|---|---|
| Global | Learnable | Align | 32 | 48 | 64 | 96 | 128 |
| ✗ | ✗ | ✗ | 5.34 | 5.32 | 5.12 | 4.99 | 4.89 |
| ✓ | ✗ | ✗ | 5.24 | 4.98 | 4.50 | 4.39 | 4.27 |
| ✓ | ✓ | ✗ | 4.17 | 3.87 | 3.43 | 3.30 | 3.21 |
| ✓ | ✓ | ✓ | **3.19** | **3.11** | **3.09** | **2.87** | **2.79** |

tokenizer (SoftVQ-B-32 (Chen et al., 2025b)) with compute control via varying diffusion steps. For our method, we use LoRA fine-tuned LightningDiT-XL models at token lengths of 32, 64, and 128 with 25 diffusion steps for quality-compute control. Our method consistently achieves better generation quality under equivalent FLOPs or throughput, surpassing both baseline VLT methods and the simple strategy of adjusting step counts with a highly compressed fixed-length tokenizer.

**System-level Comparison** Table 2 provides a system-level comparison on ImageNet $256 \times 256$ conditional generation. Because 1D tokenizers and VLT baselines differ substantially from one another in tokenizer architecture, generator family, sampling budget, etc., this table should be interpreted as a quality-compute comparison rather than a tokenizer-only ablation. We include a higher-compute variant of our method (100 sampling steps) to enable comparison in a similar gFID range. Our method achieves better gFID than existing VLTs with smaller (and simpler) tok-

enizer models, specifically without requiring a diffusion-based tokenizer decoder. Compared to conventional (fixed-length) 1D tokenizers, our method achieves competitive generation quality at high compression ratios. While a gap remains between 1D and 2D tokenizers in generation quality, exploring hybrid VLT architectures is a promising direction, which we leave for future work.

### 4.3. Analysis

We further analyze reconstruction quality, the role of each component, token similarity under global merging, and qualitative generation samples.

**Reconstruction Quality** Figure 5 shows the rFID, PSNR, and SSIM performance of image reconstruction across various token lengths. We compare the performance of our tokenizer against One-D-Piece (Miwa et al., 2025), Semanticist (Wen et al., 2025), and a tokenizer trained with nested dropout on our model architecture. Our tokenizer achieves

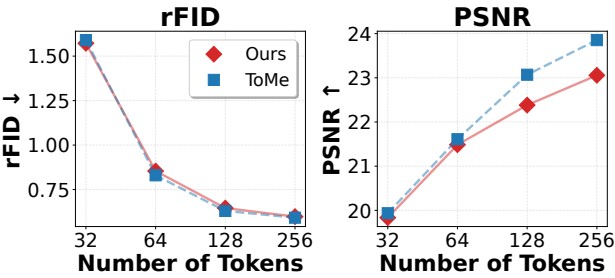

*Figure 6.* **Reconstruction comparison between LGM and ToMe-style data-dependent merging.** We report rFID and PSNR on ImageNet $256 \times 256$ across token lengths.

strong performance across all metrics—rFID, PSNR, and SSIM—especially at higher compression ratios. While Semanticist exhibits comparable rFID at 32 tokens, it uses a diffusion decoder, which hinders exact reconstruction, resulting in poor PSNR and SSIM metrics.

**Ablation Study** In Table 3, we conduct an ablation study on (1) global merging, which enables the use of positional embedding and proportional attention within DiTs (without global merging indicates conventional merging, i.e., ToMe (Bolya et al., 2023)), (2) learnable embeddings, and (3) alignment loss. The results show that the use of data-dependent merging without positional embedding and proportional attention greatly degrades the generation quality, while learnable embeddings and alignment loss consistently improve the generation quality. Notably, the ToMe-style data-dependent merging variant, which does not use global merging, leads to worse gFID across all token lengths, suggesting the importance of applying proportional attention and merged positional embeddings consistently during generation. We further compare LGM with this ToMe-style variant in terms of reconstruction quality using rFID and PSNR in Figure 6. Since ToMe selects image-specific merge pairs, it is expected to be favorable for reconstruction; however, the observed reconstruction gap is small compared with the generation-quality degradation in Table 3.

**Token Similarity under Global Merging** We further analyze whether the learned global pattern merges similar latent tokens despite being image-agnostic. Table 4 reports the average token-wise cosine similarity of all token pairs, pairs merged by LGM, and pairs merged by ToMe.[5] ToMe obtains the highest similarity because it selects merge pairs separately for each image. Nevertheless, LGM remains far above the all-pair average across token counts. For instance, at 32 tokens, LGM-merged pairs have an average similarity of 0.669 compared with 0.142 for all pairs, which supports the interpretation that the learned global pattern tends to merge similar tokens, consistent with the condition

---

[5]All similarities are measured within our trained tokenizer.

*Table 4.* **Average token-wise cosine similarity of merged token pairs.** LGM uses a learned image-agnostic pattern, while ToMe selects image-specific pairs.

| # Tokens | All pairs | LGM | ToMe |
|---|---|---|---|
| 128 | 0.142 | 0.858 | 0.906 |
| 64 | 0.142 | 0.773 | 0.805 |
| 32 | 0.142 | 0.669 | 0.707 |

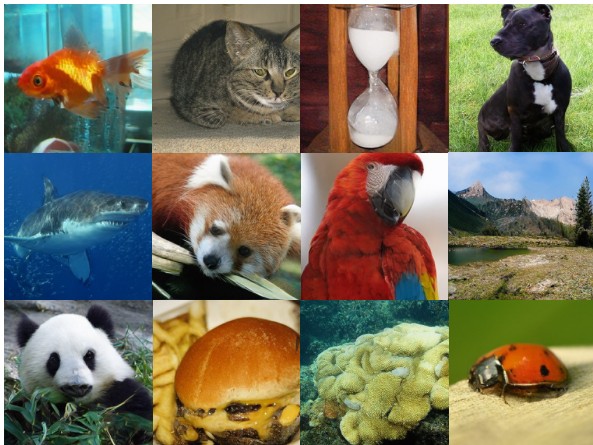

*Figure 7.* **Generated image results.** We show class-conditional generated images on ImageNet $256 \times 256$ LightningDiT-XL (+LoRA) with token length of 32.

for reducing the representation shift in Eq. 5.

**Qualitative Samples** Figure 7 provides qualitative class-conditional samples generated by our model on ImageNet $256 \times 256$. The samples illustrate that the proposed variable-length generation pipeline can produce visually coherent images even at a compact token length of 32. Additional qualitative results are provided in the Appendix.

## 5. Conclusion

We propose a variable-length tokenizer based on learnable global merging that encourages representational alignment across token lengths. By making the merging pattern data-independent, the tokenizer keeps length modulation compatible with diffusion-based generation while retaining a direct alignment objective across token lengths. This improves the training of diffusion models that operate across different token counts, enabling flexible generation of high-quality images under varying computational budgets. Our experiments show that our method encourages representational alignment while improving the generalizability of variable-length diffusion models. We evaluate our tokenizer and diffusion transformer on the ImageNet-1K dataset, demonstrating effective length modulation that achieves favorable quality-compute trade-offs compared with prior variable-length tokenization pipelines.

## Acknowledgments

This work was in part supported by the National Research Foundation of Korea (RS-2024-00351212 and RS-2024-00436165), the Institute of Information & communications Technology Planning & Evaluation (IITP) (RS-2024-00509279, RS-2022-II220926, and RS-2022-II220959, RS-2019-II190075), and the High-Performance Computing Support Project funded by the Korea government (MSIT). We thank Jaehoon Yoo for insightful discussions. We also thank Whie Jung, Kiet T. Nguyen, and Yeonwoo Cha, all at KAIST, for their help in improving the clarity of the manuscript.

## Impact Statement

This work focuses on developing a tokenizer that enables diffusion models (or possibly other generative models) to flexibly balance quality and computational cost, allowing generative models to be more efficiently deployed across diverse scenarios and environments. Since our contribution lies in improving efficiency rather than introducing new capabilities, we believe there are no ethical concerns or harmful broader impacts to highlight.

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

# A. Pseudo Code

We provide PyTorch-style pseudocode for Learnable Global Merging (LGM).

---
**Algorithm 1** Pytorch style pseudocode for LGM
---

```python
def LGM(x, embeddings, num):
    # learnable embeddings -> cosine similarity
    e = F.normalize(embeddings, dim=-1)
    S = e @ e.T

    # hard clustering (detached)
    labels = agglomerative_cluster(S, num)
    A = F.one_hot(labels, num).float()   # (N, num)
    m = A.sum(dim=0) # effective size

    # cluster prototypes
    c = F.normalize((A / A.sum(0)).T @ e, dim=-1)   # (num, D)

    # straight-through soft assignment
    A_soft = F.softmax(e @ c.T / tau, dim=-1)
    A_st = A + A_soft - A_soft.detach()

    # normalize -> merge
    W = A_st / A_st.sum(dim=0)                       # (N, num)
    z = torch.einsum('bnc, nm -> bmc', x, W)   # (B, num, C)

    return z, m
```

---

# B. More Implementation Details

For tokenizer training, we largely follow the 1D ViT tokenizer (Yu et al., 2024) design from SoftVQ (Chen et al., 2025b), with modifications to incorporate our length modulation module via learnable global merging and the alignment loss ($\mathcal{L}_{\text{align}}$). For DiT training, we follow the training recipe from LightningDiT (Yao et al., 2025) adapted for 1D tokenizers, as in MAETok (Chen et al., 2025a).

**Tokenizer Architecture**   We implement our variable-length tokenizer with the recently proposed 1D tokenizer framework (Yu et al., 2024). Unlike 2D tokenizers that enforce spatial correspondence between latent tokens and image patches, 1D tokenizers operate on 1D token sequences without enforcing spatial correspondence, thereby offering greater flexibility for high compression and variable-length tokenization. We adopt the architecture of SoftVQ (Chen et al., 2025b), which demonstrates state-of-the-art performance for highly compressed representations with 32 or 64 tokens. The architecture consists of three parts: (1) encoding, (2) length modulation, and (3) decoding.

Specifically, 1D tokenizers employ a Vision Transformer (ViT)-based encoder-decoder structure, as in (Chen et al., 2025b; Yu et al., 2024; Chen et al., 2025a). For encoding, the input images $\boldsymbol{x} \in \mathbb{R}^{H \times W \times 3}$ are first divided into non-overlapping patches and linearly projected into image patch embeddings $\boldsymbol{p} \in \mathbb{R}^{M \times d}$, where the total number of image patches $M$ is determined by the patch size $p$ and the embedding dimension $d$. Then, the patch embeddings are concatenated with $N$ learnable latent tokens $\boldsymbol{l} \in \mathbb{R}^{N \times d}$, which are introduced to aggregate information from the image patches. These $N + M$ tokens are fed into the ViT encoder $f_{\text{enc}}$, and we retain only the $N$ latent tokens, *i.e.*, $\hat{\boldsymbol{l}}$, and apply soft quantization (Chen et al., 2025b), which maps the ViT encoder output into the codebook $\mathcal{C} \in \mathbb{R}^{N \times d}$ using soft categorical distributions to enhance compression, as follows:

$$\hat{\boldsymbol{l}} = f_{\text{enc}}(\mathbf{x}; \mathbf{l}), \quad \mathbf{z} = \text{Softmax}\left(-\frac{||\hat{\boldsymbol{l}} - \mathcal{C}||_2}{\tau}\right)\mathcal{C}. \tag{15}$$

After encoding, we apply our merging-based length modulation to reduce the token count to $N - k$ on the encoder output:

$$\tilde{z} = \bar{\Gamma}^\intercal z \tag{16}$$

where the assignment matrix $\Gamma$ is derived from learnable embeddings $e$ in Eq. 10. Through this process, the encoder produces latent tokens of variable length $N - k$ within our framework.

The decoder then reconstructs the original image from the length-modulated latent tokens. For decoding, we follow the reverse process: given latent tokens $z$ or merged tokens $\tilde{z}$, learnable mask tokens $l^{\text{mask}}$ are concatenated with the latent tokens and processed by the ViT decoder to produce image patch embeddings, which are then linearly projected to reconstruct the image. When processing the merged tokens with the ViT decoder, we employ proportional attention (Bolya et al., 2023) to account for the effective size of merged tokens.

**Evaluation Protocol** For gFID, we report FID@50K with classifier-free guidance (Ho & Salimans, 2021) (CFG) where the optimal scale is chosen by FID@10K results. The gFID is computed against the whole training dataset. We also use a classifier guidance interval from 0.15 to 1. For rFID, we report FID@50K against the validation dataset. For computational cost, we measure floating-point operations (FLOPs) and throughput for the whole generation process, including detokenization. For throughput, we measure images per second on a single NVIDIA 3090 GPU with fp32.

## C. Results on 512×512 Resolution

We conduct additional experiments on ImageNet 512×512. We evaluate three aspects: reconstruction quality, representational alignment across token lengths measured by CKNNA, and conditional generation quality using LightningDiT-B, as in the main paper. For a controlled comparison, we train both our learnable global merging tokenizer and the nested dropout baseline under identical settings, following the setup of Table 1, but at 512×512 resolution.

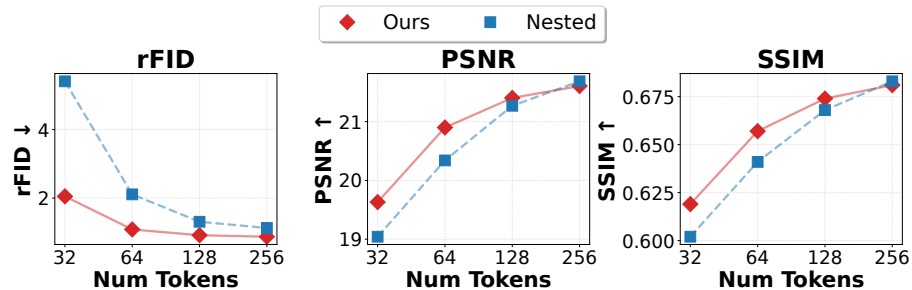

*Figure 8.* **Reconstruction quality at 512×512 resolution.**

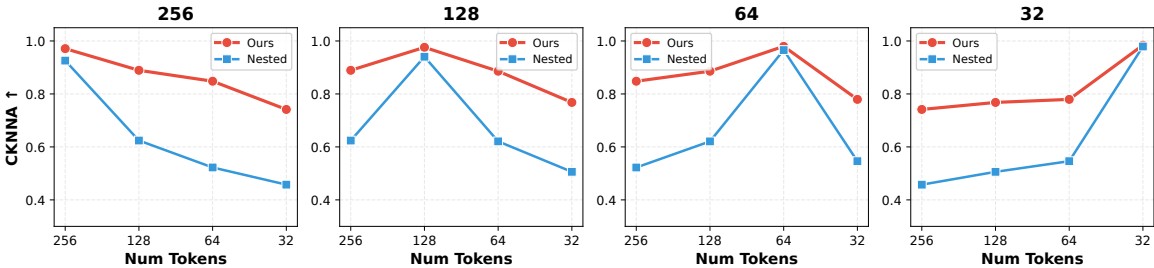

*Figure 9.* **Cross-length representational alignment at 512×512 resolution.** Higher is better.

As shown in Figures 8 and 9, the 512×512 results are consistent with the findings in the main paper. Our method provides better reconstruction quality, particularly under high compression, in terms of rFID, PSNR, and SSIM. Our method also yields higher cross-length CKNNA scores than nested dropout, indicating that learnable global merging encourages more consistent representations across token lengths.

*Table 5.* **Conditional generation at 512×512 resolution.** We report gFID of a variable-length LightningDiT-B trained jointly across token lengths on ImageNet 512×512. Lower is better.

| gFID / #Tokens | 32 | 64 | 128 |
|---|---|---|---|
| Ours | 3.99 | 3.86 | 3.81 |
| Nested dropout | 5.65 | 4.15 | 3.84 |

Table 5 further shows that our tokenizer leads to better conditional generation quality when used with a variable-length diffusion model trained jointly across token lengths. Compared with nested dropout, our method improves gFID by 1.66, 0.29, and 0.03 at 32, 64, and 128 tokens, respectively.

## D. Qualitative Results

We present additional qualitative generation results using LightningDiT-XL at 32 tokens with 100 sampling steps.

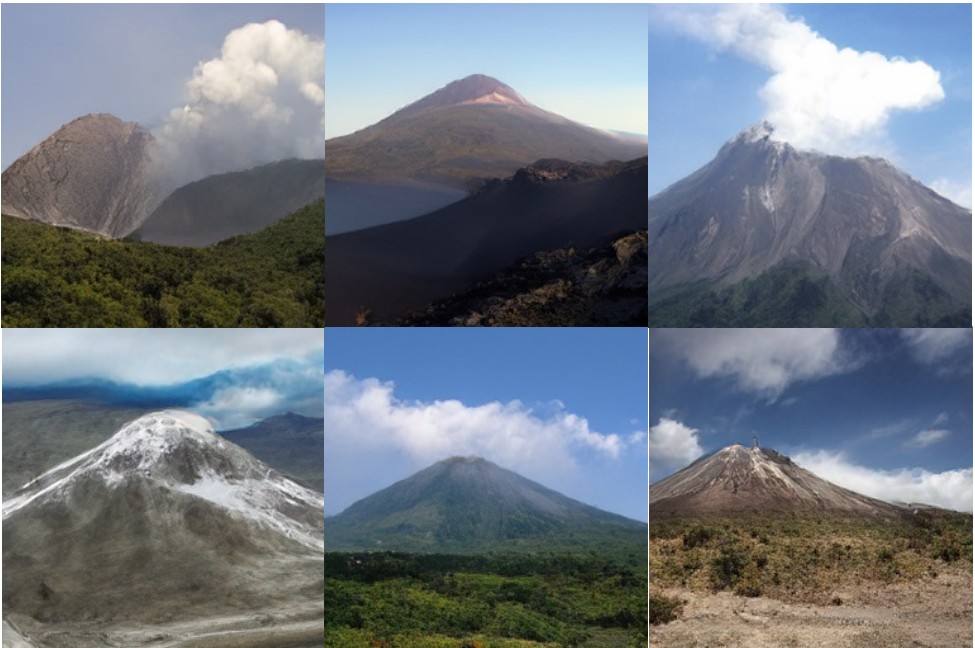

*Figure 10.* class index 980: volcano

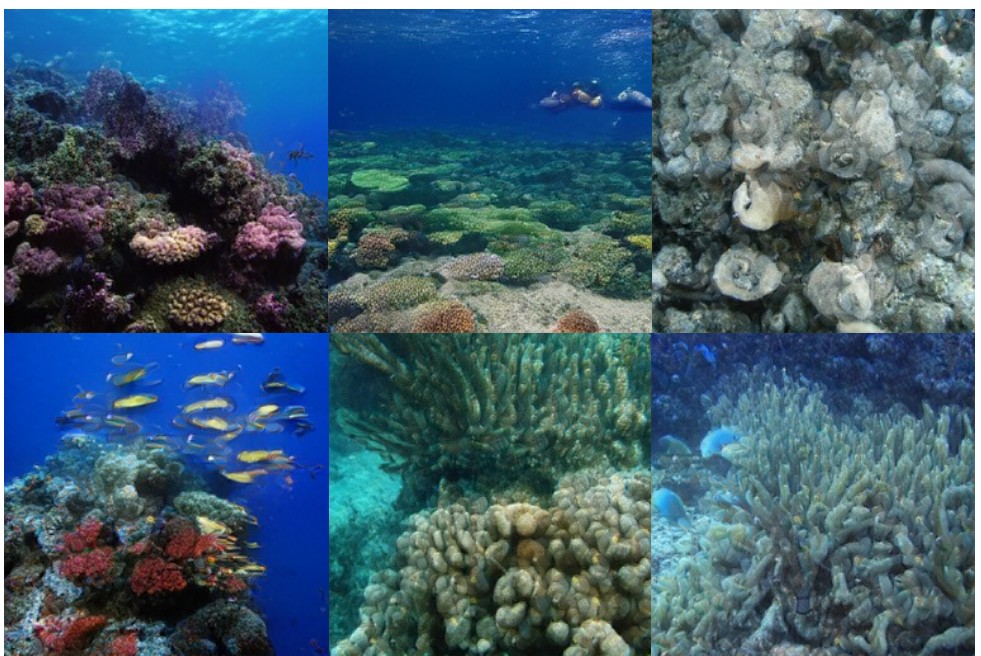

*Figure 11.* class index 973: coral reef

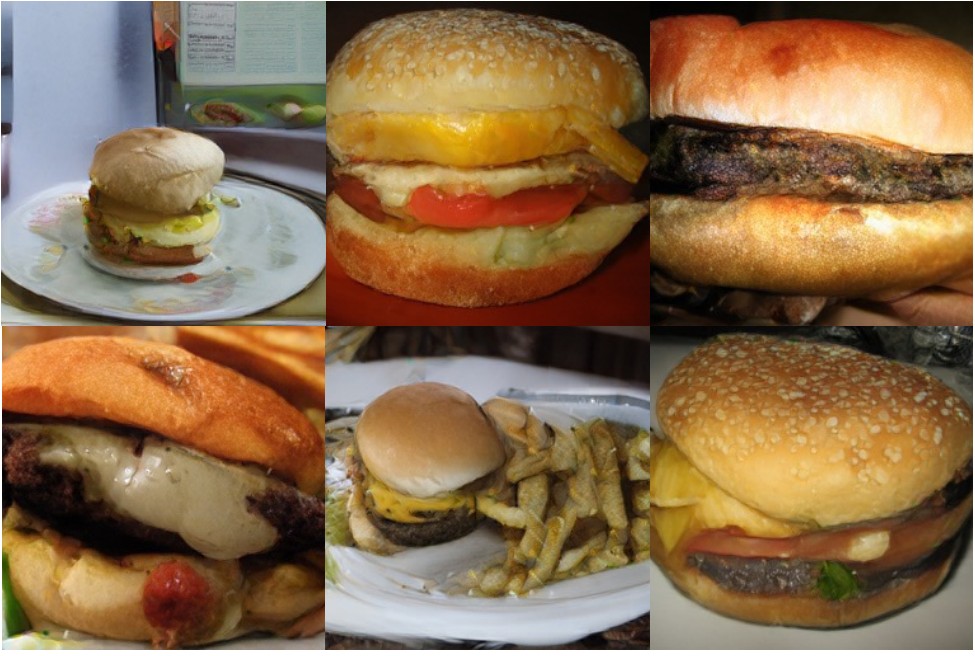

*Figure 12.* class index 933: cheeseburger

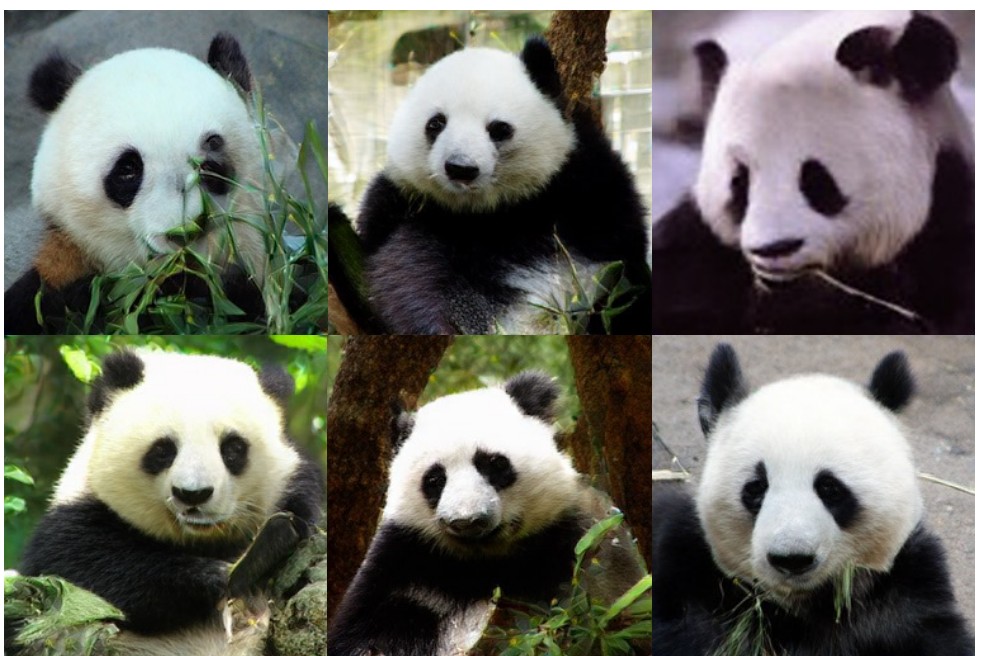

*Figure 13.* class index 388: giant panda

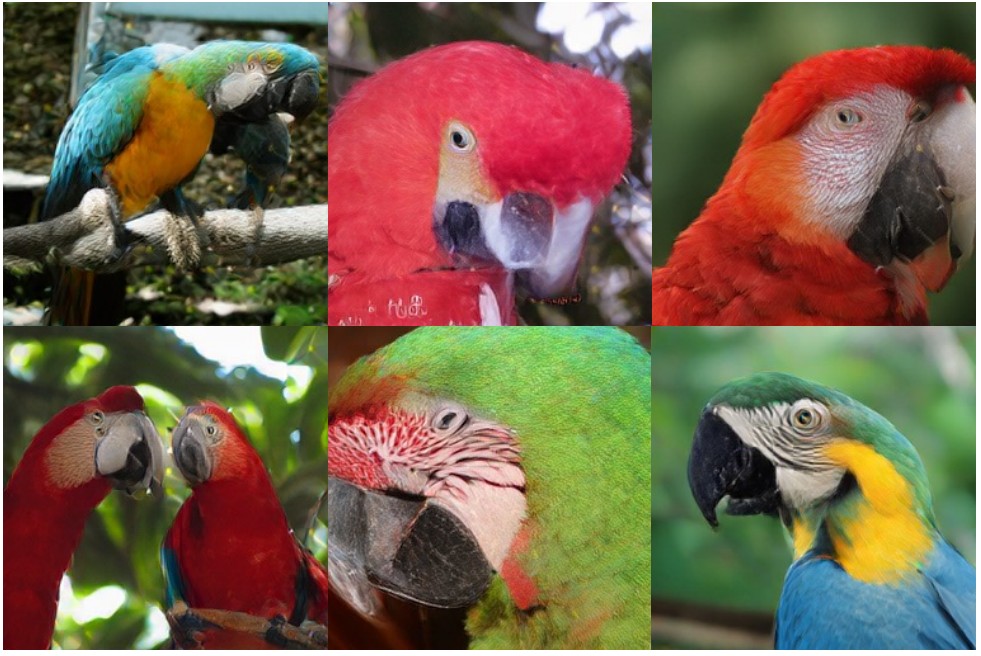

*Figure 14.* class index 88: macaw

