# OpenReview forum: "Variable-Length Tokenization via Learnable Global Merging for Diffusion Transformers"
_ICML.cc/2026/Conference — ICML 2026 regular_

### Official Review · Reviewer_eyff · 2026-03-10

**Soundness:** 3
**Presentation:** 3
**Significance:** 2
**Originality:** 3
**Overall Recommendation:** 4
**Confidence:** 4

**Summary:**

This paper proposes a variable-length tokenizer which for Diffusion Transformers. Its key idea is using a merge-based technique to enable flexible quality-compute tradeoffs. This is different from exisiting variable-length tokenizers, which often rely on nested dropout that leads to representational misalignment. More specifically, this paper propose Learnable Global Merging, which uses learnable embeddings to determine a data-independent merging pattern. This ensures the merging structure is still available during generation. Experiments on ImageNet 256 demonstrate the superiority of the proposed method, showcasing a better gFID-compute tradeoff compared to prior methods.

**Compliance With Llm Reviewing Policy:**

Affirmed.

**Final Justification:**

The rebuttal adressed my concerns and I remain my score as stated in my response.

**Key Questions For Authors:**

1. I'm curious if there are some spatial heuristics in the learned global merging patterns. Could you provide some visualizations on this?

2. How does the straight-through trick affect the training stability of the tokenizer?

3. How sensitive is the model performance to the alignment loss weight $\lambda_{align}$ and the margin $\delta$?

**Limitations:**

yes

**Strengths And Weaknesses:**

- Strengths
1. The idea of utilizing merging rather than nested dropout for handling variable-length tokenization is interesting.

2. The concept of learnable global merging effectively bridges the gap between adaptive token compression and the requirement for data-independency in generative models.

3. The use of proportional attention and merged positional embeddings provides a technically grounded way to handle variable-length inputs in DiTs.

- Weaknesses

1. The best results rely on length-specific LoRA fine-tuning, which adds a secondary training stage and complicate the overall pipeline.

2. Since merging is data-independent, the fixed pattern might not optimally capture unique local structures in highly irregular images.

---

> ### Author Rebuttal · Authors · 2026-03-31
>
> We thank the reviewer for valuable comments. Below we clarify individual concerns.
>
> > W1. The additional complexity from length-specific LoRA fine-tuning.
>
> As the reviewer pointed out, length-specific LoRA fine-tuning introduces an additional training stage.
> However, we clarify that the cost is minimal relative to the performance gain, and our overall (training/inference) pipeline remains substantially simpler than those of other variable-length tokenizers.
>
> In terms of training, LoRA fine-tuning achieves performance comparable to separately trained models (within -0.2 to +0.5 gFID on DiT-B) with only 2.5% additional training per length (10 epochs vs. 400 epochs for full training), enabling highly efficient post training. At inference, length-specific LoRA adds only 2.4~3.4% additional parameters, requiring minimal extra memory cost.
>
> Importantly, compared to other variable-length tokenizers, our pipeline is considerably simpler.
> For instance, Semanticist and FlexTok require a diffusion-based decoder, making their tokenizer training itself a two-stage process. At inference time, these methods also require two sequential generative models (e.g., AR + diffusion), whereas our method uses a single diffusion model with a deterministic decoder.
> Given this, the addition of lightweight LoRA fine-tuning introduces less complexity than the multi-stage pipelines of existing approaches.
>
> > W2. The fixed pattern might not optimally capture unique local structures.
>
> We acknowledge that a fixed merging pattern is not the optimal solution for minimizing Eq. 5 or maximizing reconstruction quality. However, data-dependent merging methods (e.g., ToMe) produce image-specific merging patterns unavailable at generation time, preventing the use of proper positional embeddings (PE) and proportional attention (PA). Since our work focuses on variable-length tokenization for image generation, we choose data-independent merging to ensure generation compatibility.
>
> To justify this choice, we provide a detailed comparison between our learnable global merging (LGM) and data-dependent merging (ToMe) in our response to Reviewer gxuo-L1. In summary, LGM achieves substantially higher within-cluster similarity than the dataset average (e.g., 0.669 vs. 0.142 at 32 tokens) despite being data-independent, and the reconstruction gap between LGM and ToMe is marginal (rFID within 0.02, PSNR within 0.8 dB), while the generation gap from incompatibility of data-dependent merging with diffusion transformers is substantially larger (gFID ~2.1 on average, Table 3).
>
> > Q1. Visualizing merging patterns.
>
> We provide a Sankey diagram illustrating how 256 tokens are progressively merged, along with an effective size histogram at 32 tokens. The learned merging patterns do not exhibit simple heuristics such as naive subsampling or single dominant clusters. Instead, tokens display a diverse range of merging sensitivity, from aggressively merged large clusters to tokens that remain unmerged even at 32 tokens.
>
> * Sankey - https://anonymous.4open.science/r/LGM-518D/sankey.png
> * Histogram - https://anonymous.4open.science/r/LGM-518D/32_bar_plot.png
>
> > Q2. Effect of the straight-through trick on training stability?
>
> We provide training loss curves of our tokenizer with the straight-through trick under different configurations (for sensitivity analysis in answer [Q3]).
>
> * Loss curve - https://anonymous.4open.science/r/LGM-518D/loss_curves.png
>
> The training loss decreases smoothly without divergence throughout training, indicating that the straight-through trick does not introduce substantial instability.
>
> > Q3. Sensitivity of the model performance to $\lambda_{align}$ and $\delta$
>
> Following the reviewer's comment, we conducted experiments on sensitivity to $\delta$ and $\lambda_{\text{align}}$, reporting reconstruction performance and average CKNNA (between 256 and 32, 64, 128 tokens).
>
> For $\delta \in \{0, 0.1, 0.25\}$ and $\lambda_{\text{align}} \in \{0, 0.1, 0.4\}$, the reconstruction differences are marginal across all settings (within 0.24 PSNR, 0.007 SSIM, 0.005 LPIPS, and 0.10 rFID), as shown in the links below.
>
> * Reconstruction ($\delta$) - https://anonymous.4open.science/r/LGM-518D/reconstruction_delta.png
> * Reconstruction ($\lambda$) - https://anonymous.4open.science/r/LGM-518D/reconstruction_lambda.png
>
> In terms of CKNNA, $\delta$ and $\lambda_{\text{align}}$ have a more noticeable effect:
>
> | $\delta$ | Avg CKNNA$\uparrow$ | | $\lambda_{\text{align}}$ | Avg CKNNA$\uparrow$ |
> |:---:|:---:|:---:|:---:|:---:|
> | 0 | 0.699 | | 0 | 0.694 |
> | 0.1 | 0.702 | | 0.1 (ours) | 0.730 |
> | 0.25 (ours) | 0.730 | | 0.4 | 0.711 |
>
> These results indicate that reconstruction quality is largely insensitive to the choice of $\delta$ and $\lambda_{\text{align}}$, while representational alignment (CKNNA) is moderately affected, confirming that these hyperparameters primarily control the degree of cross-length alignment without compromising reconstruction.

---

> > ### Author Rebuttal · Reviewer_eyff · 2026-04-01
> >
> > I appreciate the clarification on the minimal cost of LoRA fine-tuning and the fair comparison with multi-stage pipelines in Semanticist/FlexTok. However, I note that maintaining separate LoRA modules for each target length introduces combinatorial complexity that those methods avoid, as they support arbitrary lengths with a single model. Please correct me if I am mistaken.

---

> > > ### Author Response · Authors · 2026-04-01
> > >
> > > We appreciate the reviewer's follow-up.
> > >
> > > We would like to clarify that our base method (without LoRA) already supports various target lengths within a single model (without length-specific parameters) as in Semanticist and FlexTok. As shown in Table 1, our merging-based length modulation without LoRA already achieves substantially better gFID than nested dropout under joint training with the same tokenizer and DiT architecture. LoRA is an optional enhancement that further improves quality at specific target lengths, not a requirement for multi-length support.
> > >
> > > While we acknowledge that the best results are obtained via LoRA fine-tuning, when LoRA is used, the additional cost scales linearly (not combinatorially) with the number of target lengths. Importantly, this merely complicates the overall pipeline, as the linear increase is marginal in both training cost (2.5% of full training epochs) and parameter count (2.4–3.4% per length), as discussed in our previous response. At inference, swapping lightweight LoRA adapters is a well-established practice in the diffusion model community, requiring minimal implementation effort (e.g., a single API call such as load_lora_weights in Hugging Face Diffusers) and incurring negligible overhead beyond a small memory cost for each length-specific LoRA parameter.
> > >
> > > Although the cost can scale if one aims to support all available lengths as targets, from a practical quality-compute trade-off perspective, the number of target lengths does not need to be large. For instance, FlexTok, despite not requiring length-specific parameters, is also trained and evaluated on a fixed set of discrete lengths (N = 1, 2, 4, 8, 16, 32 — six lengths). At a comparable number of target lengths, our total additional parameter and training cost from LoRA remains substantially smaller than the overhead introduced by their diffusion-based decoders.

---

### Official Review · Reviewer_xBwJ · 2026-03-11

**Soundness:** 3
**Presentation:** 3
**Significance:** 2
**Originality:** 2
**Overall Recommendation:** 4
**Confidence:** 4

**Summary:**

This paper argues that existing variable-length tokenizers based on nested dropout are poorly suited for a single diffusion model operating across multiple token lengths, because truncation changes token semantics and disrupts cross-length representational alignment. To address this, the authors propose a merging-based variable-length tokenizer in which token count is reduced by merging tokens rather than dropping suffix tokens, together with a learnable global merging mechanism that is data-independent and therefore compatible with generation-time diffusion transformers. The paper further adapts DiT-style models with proportional attention and merged positional embeddings, and reports improved CKNNA alignment scores and better gFID-compute trade-offs on ImageNet 256x256 relative to prior variable-length tokenization baselines. The paper also explores a lightweight length-specific LoRA post-training step to further refine performance.

**Compliance With Llm Reviewing Policy:**

Affirmed.

**Final Justification:**

After reading all the authors’ and other reviewers’ comments, I decide to raise my score to 4.

**Key Questions For Authors:**

- Do you have any controlled analysis of how far learnable global merging is from a data-dependent merging upper bound, at least for tokenizer reconstruction or for an oracle setting? That would help assess the price of the generation-compatibility compromise.
- Can you clarify the sampling-step setup across Figure 4 and Table 2? The paper seems to use different step counts for the proposed method in different places. I need a precise explanation of which comparisons are matched in FLOPs and which are not.

**Limitations:**

yes

**Strengths And Weaknesses:**

### Strengths:
- In general this paper is well-written and easy to follow.
- This paper tackles an important problem for latent learning, namely how to support multiple compute budgets without training a separate tokenizer and generative model for each compression rate.
- The system-level numbers are at least promising. In Table 2, the 32-token version of the proposed method achieves much higher throughput than the compared variable-length baselines while staying in a similar gFID range, which is practically relevant if the goal is flexible deployment.

### Weaknesses:
- The main system-level comparisons are too confounded to cleanly attribute gains to the tokenizer. In Table 2, the paper compares against systems using different tokenizer classes, different generator families, different parameter budgets, and different sampling configurations. If the paper wants to claim that the proposed tokenization principle is better, it needs more same-backbone or same-generator controls.
- The scope of empirical validation is limited. All experiments are on ImageNet 256x256. For a systems-and-representation paper making fairly broad claims about variable-length tokenization for diffusion transformers, that feels thin. A second dataset, a second resolution, or even an unconditional setting would help determine whether the learned global merging pattern generalizes beyond this one benchmark.
- The proposed global merging is data-agnostic and determined solely by position embeddings rather than image content. While this makes the method compatible with generation, it also forces a fixed merging pattern across all images, which may only capture dataset-level statistics instead of image-specific token similarity. This partially contradicts the theoretical motivation of merging similar tokens and may limit the achievable alignment and compression efficiency.

---

> ### Author Rebuttal · Authors · 2026-03-31
>
> We thank the reviewer for valuable comments.
>
> > W1. The main system-level comparisons are too confounded (Tab 2)
>
> We respectively remind the reviewer that the controlled comparison the reviewer requests is provided in Tab 1, where we use the same tokenizer architecture (ViT-B), generative model (LightningDiT-B), and sampling strategy (ODE, 250 steps), clearly demonstrating the advantage of our learnable global merging over nested dropout under identical conditions.
>
> Tab 2 is intended to provide "system-level" comparisons to existing tokenization methods. While we acknowledge that the comparison involves heterogeneous configurations, this is largely unavoidable given the vastly different design choices entangled with their tokenizers. For instance, FlexTok and Semanticist employ diffusion-based decoders and AR generators as the core design components, making it difficult to align either the tokenizer or generator class. Since it is not straightforward nor fair to forcefully match the settings, we use the Tab 2 to provide the system-level comparisons to their best-achievable settings, showcasing that our framework achieves slightly lower peak performance compared to separately trained fixed-length 1D tokenizers while offering a superior quality-compute trade-off over existing baselines.
>
> > W2. Broadening the empirical validation scope.
>
> To broaden the empirical validation, we conduct more experiments at 512 resolution, evaluating reconstruction quality, representational alignment (CKNNA), and conditional generation with DiT-B.\
> (please refer to our response to Reviewer 3xHD-W1 for full details)
>
> Our method consistently provides (1) better reconstruction quality in the high compression regime (rFID: 5.4→2.1 at 32 tokens), (2) substantially higher cross-length CKNNA scores (0.82 vs. 0.55 on average), and (3) better generation quality (1.66 gFID improvement at 32 tokens), showing that our method scales effectively to higher resolutions.
>
> We note that our method does not involve any dataset-or resolution-dependent components and is built on a standard 1D ViT tokenizer, so we believe there are no fundamental barriers to scaling to other datasets or resolutions.
>
> > W3/Q1 LGM may limit the achievable alignment and compression.
>
> We respectfully argue that LGM does not contradict the motivation of merging similar tokens. Our motivation establishes that merging similar tokens reduces representation shift. LGM is a generation-compatible realization of this by learning embeddings that reflect dataset-level token similarity patterns and training them jointly with the alignment loss to encourage similar tokens to merge.
>
> To support this, in our response to Reviewer gxuo-L1, we provide a comparison of within-cluster cosine similarity and reconstruction quality between LGM and data-dependent merging (ToMe). The analysis confirms that LGM achieves substantially higher within-cluster similarity than the dataset average (0.669 vs. 0.142 at 32 tokens), and the reconstruction gap is marginal (~0.02 rFID, ~0.8 PSNR). Importantly, this reconstruction gap is far smaller than the generation gap (gFID ~2.1 on avg) caused by the incompatibility of data-dependent merging with DiTs (Tab 3).
>
> In summary, we agree that data-dependent merging would minimize Eq. 5 more directly and can provide better reconstruction. However, we choose LGM for generation compatibility, since such methods produce image-specific merging patterns unavailable at generation time, preventing the use of proper positional embeddings and proportional attention. The resulting generation gap is substantial compared to the reconstruction gap from adopting LGM.
>
> > Q2. Clarification on Fig 4 and Tab 2.
>
> We apologize for the confusion.
>
> In Fig 4, we use 25 sampling steps for our method, as 25 steps provide a good balance between inference speed and quality (as in FlexTok). This comparison shows that our method achieves a better quality-compute trade-off within the computational cost and gFID range covered by Semanticist-L, FlexTok-d18, and SoftVQ-B-32.
>
> In Tab 2, we also compare against larger and higher-performing VLT models, such as FlexTok-d28 and Semanticist-XL, scaled-up versions of FlexTok-d18 and Semanticist-L. While these models operate in a different gFID and cost range, we observed that our method with 100 sampling steps reaches a similar gFID range, enabling a direct quality-compute comparison. Notably, our method achieves comparable gFID with substantially lower computational cost and model sizes, highlighting the efficiency of our approach without requiring larger models.
>
> We acknowledge that this difference in steps across Fig 4 and Tab 2 was not clearly explained. In the revised version, we will present both 25-step and 100-step results in Tab 2 (https://anonymous.4open.science/r/LGM-518D/table_2.png) and clarify that the 100-step configuration serves as a higher-compute variant of our pipeline for comparison against these larger-scale baselines.

---

> > ### Author Rebuttal · Reviewer_xBwJ · 2026-04-03
> >
> > Thank you for your rebuttal. After reading all the authors’ and other reviewers’ comments, I decide to raise my score to 4.

---

> > > ### Author Response · Authors · 2026-04-03
> > >
> > > We are very glad to hear that your concerns have been addressed. \
> > > All the comments and discussions raised during the review will be included in the revised version of our paper. \
> > > We thank again for the thoughtful review and for taking the time to consider our rebuttal along with the other reviewers’ comments.

---

### Official Review · Reviewer_gxuo · 2026-03-12

**Soundness:** 2
**Presentation:** 2
**Significance:** 2
**Originality:** 2
**Overall Recommendation:** 4
**Confidence:** 3

**Summary:**

This paper proposes a variable-length tokenizer for diffusion transformers. Instead of shortening latents by truncating tokens, it merges similar tokens to better preserve semantics across different token counts. It learns an image-agnostic global merging policy so the same rule can be used at generation time, then trains one DiT to work across multiple lengths. On ImageNet 256×256, it reports better cross-length consistency and a stronger quality–compute trade-off than prior variable-length tokenizers.

**Compliance With Llm Reviewing Policy:**

Affirmed.

**Key Questions For Authors:**

- The method jointly optimizes the merging embeddings together with the tokenizer encoder and decoder, rather than learning only an image-agnostic merging policy. What is the necessity of this joint optimization? In particular, how much performance would be lost if the encoder/decoder were kept frozen and only the merging module were trained?

**Limitations:**

- The paper introduces an image-agnostic global merging module as a practical regularization for generation-time compatibility, but this design departs from the paper’s ideal alignment formulation, under which clustering the actual latent tokens would more directly minimize representation shift. A more explicit analysis comparing latent-dependent merging and the proposed image-agnostic surrogate would strengthen the claim.

- It is not clear that merging is the only viable design choice for variable-length diffusion. The paper does not compare against a coarse-to-fine splitting alternative (e.g., expanding from 32 tokens), so the advantage of merging over other structured length-modulation strategies remains unverified.

**Strengths And Weaknesses:**

- The paper identifies a real limitation of truncation-based variable-length tokenizers: they hurt cross-length representational alignment, which makes a single diffusion model hard to train across multiple token counts.

- Intensive experiments and ablation are conducted to verify its each component.

---

> ### Author Rebuttal · Authors · 2026-03-31
>
> We thank the reviewer for valuable comments. Below we clarify individual concerns.
>
> > Q1. Necessity of joint optimization.
>
> The joint optimization of the merging policy together with the tokenizer (encoder and decoder) is essential for our method. In terms of the encoder and decoder training, joint optimization enforces the latent tokens learned by the tokenizer to be merging-friendly, i.e., under a given merging policy, the latent tokens receive gradients from the reconstruction loss, making degradation by merging less likely. Without joint optimization, variable-length tokenization with minimal information loss is difficult to achieve and incurs significant reconstruction degradation.
>
> To verify this, we conduct the experiment the reviewer suggested: we freeze a pretrained tokenizer (trained without merging) and separately train our learnable global merging (Ours) or apply data-dependent merging at inference time (ToMe).
> As shown in the table below, without joint optimization, reconstruction quality degrades drastically even for data-dependent merging (ToMe) which selects the most similar token pairs.
> This indicates that a pretrained encoder without merging-aware training produces latents where even the most similar tokens carry sufficiently distinct information that merging them causes significant information loss.
>
> We also note that fixing the merging policy and training only the tokenizer also leads to generation degradation, as shown in Table 3 (row without "Learnable"), confirming that both components benefit from joint optimization.
>
> |   rFID   |  256 |  128  |   64   |   32  |
> |:--------:|:----:|:-----:|:------:|:-----:|
> | Baseline | 0.46 |   -   |    -   |   -   |
> |   Ours   | 0.46 | 74.17 | 152.83 | 177.6 |
> |   ToMe   | 0.46 | 27.17 |  111.4 | 139.5 |
>
> |   PSNR   |  256  |  128  |   64  |   32  |
> |:--------:|:-----:|:-----:|:-----:|:-----:|
> | Baseline | 25.93 |   -   |   -   |   -   |
> |   Ours   | 25.93 | 16.53 | 13.44 | 12.89 |
> |   ToMe   | 25.93 |  18.6 | 15.91 | 14.64 |
>
> > L1. A more explicit analysis comparing latent-dependent/-agnostic surrogate.
>
> Following the reviewer's comment, we provide more analysis comparing our data independent learnable global merging (LGM) and data-dependent merging (ToMe).
>
> We first compare the average cosine similarity between (a) all token pairs, (b) tokens merged by LGM, and (c) tokens merged by ToMe:
>
> | cos similarity | inter  | ours   | tome   |
> |:--------------:|:------:|:------:|:------:|
> |      128       | 0.1418 | 0.8576 | 0.9056 |
> |       64       | 0.1418 | 0.7733 | 0.8053 |
> |       32       | 0.1418 | 0.6686 | 0.7074 |
>
> The results show that while our global merging does not reach the similarity level of data-dependent merging, it achieves substantially higher within-cluster similarity than the dataset average (e.g., 0.669 vs. 0.142 at 32 tokens).
> This confirms that although our method is not data-dependent, it encourages merging similar tokens thereby reducing the representation shift in Eq.5.
>
> We also compare reconstruction quality across token lengths:
>
> * Comparison - https://anonymous.4open.science/r/LGM-518D/ours_vs_tome.png
>
> The reconstruction gap is marginal compared to the generation gap: the rFID difference is within 0.02 across all token counts, with PSNR gaps ranging from 0.1 to 0.8 dB, while the generation gap caused by the incompatibility of data-dependent merging with diffusion transformers is substantially larger (gFID ~2.1 on average), as shown in Table 3.
>
> (* Note that the reconstruction results are obtained with a reduced training configuration (smaller batch size and fewer epochs for both LGM and ToMe) due to the limited rebuttal period, which may cause slight differences from the numbers reported in the paper.)
>
> > L2. Comparison against a coarse-to-fine splitting alternative
>
> We understand the reviewer's suggestion as referring to adaptive-length tokenizers such as ALIT, which expand tokens in a coarse-to-fine manner.
> We note that these methods perform instance-dependent compression, where the compression ratio varies per image, and are primarily designed for efficient recognition rather than generation.
>
> Adapting such methods for generation is not straightforward: since the compression ratio is determined per instance, there is no principled way to decide token allocation during denoising (where no input image is available), risking out-of-distribution decoding outside proper compression ratio. Additionally, as with instance-dependent merging, positional embeddings become challenging since the token structure varies per instance. Indeed, these works [ALIT, AIT] do not include experiments on generation tasks.
>
> Making these methods generation-compatible and comparing quality-compute trade-offs is an interesting direction, but we consider it out of scope for this work.
>
> [ALIT] Adaptive Length Image Tokenization via Recurrent Allocation \
> [AIT] Single-pass Adaptive Image Tokenization for Minimum Program Search

---

> > ### Author Rebuttal · Reviewer_gxuo · 2026-04-05
> >
> > Thanks for the reviewer's detailed explanation, i will keep my score.

---

> > > ### Author Response · Authors · 2026-04-06
> > >
> > > We are glad that our responses have resolved your concerns. \
> > > All suggestions and discussions throughout the rebuttal period will be included in the revised version of our paper. \
> > > We sincerely thank you for the constructive and thorough review, and for the time and effort for the discussion.

---

### Official Review · Reviewer_3xHd · 2026-03-13

**Soundness:** 3
**Presentation:** 2
**Significance:** 3
**Originality:** 4
**Overall Recommendation:** 4
**Confidence:** 5

**Summary:**

This paper presents the variable-length image tokenization method for diffusion models. Specifically, it proposes the novel merging-based approach that controls the token length while encouraging the preservation of the semantics which utilizes the mixture of cleaver strategies from old books (like, straight-through trick). At last, the paper compares against the baselines on class-conditioned image generation.

**Compliance With Llm Reviewing Policy:**

Affirmed.

**Key Questions For Authors:**

- Can authors provide the results on the 512x512 resolution tasks? At least the rFID and semantics metric?
- What is the performance of LightningDiT with the proposed tokenizer and with RoPE (or baseline absolute leanable PE without any trick) embeddings?
- Please also provide the pseudo code for better readability.

**Limitations:**

Please see the weaknesses.

**Strengths And Weaknesses:**

**Strengths**

- THis paper focuses on a very important problem. Specifically, compressing 1d VAE tokens does not lead to the improvement in DiT models but actually starts hurting the performance.
- The paper presents a novel solution that encourages the semantic features without requiring the pretrained semantic encoders.
- The results show the decent performance on gFID by training the DiT (w/ improved position embedding trick to adapt the proposed tokenizer).


**Weaknesses**

While the results are great on class-conditioned generation tasks, there are several weakness:
- It is not clear how well this method can be scaled. Specifically, if we increase the resolutions to 512x512, can we get similar rFID vs. semantics vs. gFID performance or are there other trade-offs?
- The position embedding trick for DiTs only works with the fixed resolution. RoPE like embeddings are required to scale the DiTs across the resolution. Hence, it is not clear how this can be applied to RoPE embeddings.
- Writing needs to be improved to make it easier for the readers. Specifically, adding a pseudo-code of the token-merging will help greatly to readability.
- Although few 1D tokenizers are added as baselines. For completion, it needs the comprehensive comparisons across the all tokenizers to bring everything into perspective even if this is appendix.

---

> ### Author Rebuttal · Authors · 2026-03-31
>
> We thank the reviewer for valuable comments. Below we clarify individual concerns.
>
> > W1/Q1. Results on the 512x512 resolution tasks.
>
> To address the reviewer's concern regarding scalability to higher resolutions, we conduct additional experiments on ImageNet 512×512, evaluating (1) reconstruction quality, (2) representational alignment (CKNNA), and (3) conditional generation with DiT-B.
> Specifically, we train both our learnable global merging tokenizer and a nested dropout baseline under identical settings (as in Table 1) at 512×512 resolution.
>
> - Reconstruction - https://anonymous.4open.science/r/LGM-518D/reconstruction_512.png
> - CKNNA - https://anonymous.4open.science/r/LGM-518D/cknna_512.png
>
> Consistent with the results in the paper, the results show that our method consistently provides (1) better reconstruction quality, particularly in the high compression regime, in terms of rFID, SSIM, and LPIPS (e.g., rFID: 5.4 -> 2.1 and LPIPS: 0.39 -> 0.36 at 32 tokens), and (2) substantially higher cross-length CKNNA scores compared to nested dropout (0.82 vs. 0.55 on average), confirming that our method effectively encourages consistent representations across resolutions.
>
> Also for generation, our method consistently provides better generation quality with a variable-length diffusion model trained jointly across token lengths.
> Specifically, our method consistently outperforms the nested dropout tokenizer by 1.66~0.03 gFID scores across token lengths.
>
> | gFID/#Tokens |  32  |  64  |  128 |
> |:------------:|:----:|:----:|:----:|
> |     Ours     | 3.99 | 3.86 | 3.81 |
> |    Nested    | 5.65 | 4.15 | 3.84 |
>
> Regarding the scalability, we note that our method does not involve any dataset-specific or resolution-dependent components in design, and is built on top of a standard 1D ViT tokenizer architecture. Thus, we believe there are no fundamental barriers to scaling to other datasets or resolutions. Our 512×512 experiments empirically support this. In the main paper, we focused on ImageNet 256×256 as it is the common setting adopted by existing VLT baselines (e.g., Semanticist, One-D-Piece).
>
> > W2/Q2. Applicability of RoPE
>
> We would like to clarify that our core contribution is orthogonal to the choice of positional embeddings (PEs). Eq. (14) simply aggregates the PEs of individual tokens that are merged together to inject positional information into the merged tokens. Thus, our method is not dependent on a specific type of PE. Note that we use standard learnable PEs in our experiments, following the convention of most existing 1D tokenizers, e.g., SoftVQ, MAETok.
>
> We note that applying RoPE on DiTs may not be suitable for 1D tokenizers.
> Unlike language or vision data, where relative positional relationships carry meaningful structural information, 1D latent tokens do not necessarily exhibit such properties (tokens at similar relative distances do not necessarily share similar relationships).
> Thus, most existing 1D tokenizers (SoftVQ, MAETok, etc.) adopt learnable positional embeddings (LPE) rather than RoPE, and we follow this convention.
>
> Regarding multi-resolution support, we agree that extending our method to handle multiple image resolutions would be valuable. However, in this work, we focus on a different important problem of variable-length tokenization for diffusion models and quality-compute trade-off, while being consistent with existing (variable-length) tokenizers that operate at a fixed resolution. We leave extending to multiple resolutions as an important future direction.
>
> > W3. Writing needs to be improved and pseudo-code for better readability.
>
> We agree that including pseudo-code would greatly improve readability.
> In the revised version, we will include pseudo-code for our learnable global merging procedure and improve the writing for better understanding.
>
> * Pseudo-code - https://anonymous.4open.science/r/LGM-518D/psuedo_code.png
>
> > W4. Comprehensive comparisons across all tokenizers bringing everything into perspective.
>
> We respectfully note that Table 2 was designed as a system-level comparison covering 2D tokenizers (SD-VAE, VA-VAE), 1D fixed-length tokenizers (SoftVQ, MAETok), and variable-length tokenizers (Semanticist, FlexTok, Ours) to provide a unified perspective across rFID, gFID, parameters, and computational costs.
> Since different tokenizers, especially 1D ones, often employ vastly different backbones and generative models, we focused on comparing strong representative tokenizers that either belong to the same category (variable-length tokenizers) or share the same backbone (SoftVQ, MAETok), while excluding some baselines with substantially different designs (e.g., pipelines without a continuous diffusion model).
>
> Nonetheless, we acknowledge that the coverage of 1D tokenizers could be more comprehensive. Following the reviewer's suggestion,  we will include comparisons with additional 1D tokenizers, such as TiTok and One-D-Piece, in Table 2 for completeness.

---

> > ### Author Rebuttal · Reviewer_3xHd · 2026-04-04
> >
> > As my questions/concerns are addressed, I will increase my score.

---

> > > ### Author Response · Authors · 2026-04-05
> > >
> > > We are glad that our responses have addressed your concerns. \
> > > All suggested improvements discussed during the rebuttal will be incorporated into the revised version of our paper.
> > >
> > > We thank you once again for the constructive review and for the time spent considering our rebuttal.

---

### Decision · Program_Chairs · 2026-04-30

**Decision:**

Accept (regular)

**Comment:**

This paper proposes a merging-based variable-length tokenizer for diffusion transformers, replacing the conventional nested dropout (truncation) approach with a learnable global merging (LGM) mechanism. The key insight is that merging similar tokens—rather than truncating ordered token sequences—preserves cross-length representational alignment, enabling a single diffusion transformer to operate effectively across multiple token lengths.

All four reviewers converged on a score of 4 (Weak Accept), with two reviewers (3xHd, xBwJ) raising their scores after the rebuttal. The reviewers consistently identified the problem as important and the merging-based approach as a sensible direction. However, the novelty is incremental, the evaluation scope is narrow, and the core design choice (data-independent merging) is pragmatically rather than principally motivated.

Given that all reviewers lean toward weak acceptance with no critical flaws identified, the AC recommends acceptance. Congratulations! Please keep in mind that the paper would benefit from broader evaluation and clearer articulation of the novelty in the final version.